# Multidimensionality of tree communities structure host-parasitoid networks and their phylogenetic composition

Ming-Qiang Wang[1,2†], Shi-Kun Guo[1,3†], Peng-Fei Guo[1,4†], Juan-Juan Yang[1,3], Guo-Ai Chen[1,3], Douglas Chesters[1,5], Michael C Orr[1,6], Ze-Qing Niu[1], Michael Staab[7], Jing-Ting Chen[1,3], Yi Li[8], Qing-Song Zhou[1,5], Felix Fornoff[9], Xiaoyu Shi[1], Shan Li[8], Massimo Martini[9], Alexandra-Maria Klein[9], Andreas Schuldt[10], Xiaojuan Liu[8], Keping Ma[5,8], Helge Bruelheide[11], Arong Luo[1,5,12]*, Chao-Dong Zhu[1,3,5,12,13]*

[1]CAS Key Laboratory of Zoological Systematics and Evolution, Institute of Zoology, Chinese Academy of Sciences, Beijing, China; [2]CAS Key Laboratory of Mountain Ecological Restoration and Bioresource Utilization & Biodiversity Conservation Key Laboratory of Sichuan Province, Chengdu Institute of Biology, Chinese Academy of Sciences, Chengdu, China; [3]College of Biological Sciences, University of Chinese Academy of Sciences, Beijing, China; [4]College of Pharmacy, Guizhou University of Traditional Chinese Medicine, Guiyang, China; [5]International College, University of Chinese Academy of Sciences, Beijing, China; [6]Entomologie, Staatliches Museum für Naturkunde Stuttgart, Stuttgart, Germany; [7]Ecological Networks, Technical University Darmstadt, Darmstadt, Germany; [8]State Key Laboratory of Vegetation and Environmental Change, Institute of Botany, Chinese Academy of Sciences, Beijing, China; [9]Department of Nature Conservation and Landscape Ecology, Albert-Ludwigs-University Freiburg, Freiburg, Germany; [10]Forest Nature Conservation, University of Göttingen, Göttingen, Germany; [11]Institute of Biology/Geobotany and Botanical Garden, Martin Luther University Halle-Wittenberg, Halle, Germany; [12]National Key Laboratory of Animal Biodiversity Conservation and Integrated Pest Management, Institute of Zoology, Chinese Academy of Sciences, Beijing, China; [13]State Key Laboratory of Integrated Pest Management, Institute of Zoology, Chinese Academy of Sciences, Beijing, China

**\*For correspondence:**
luoar@ioz.ac.cn (AL);
zhucd@ioz.ac.cn (C-DZ)

†These authors contributed equally to this work

## eLife Assessment

This **valuable** study uses a massive and long-term experimental data set to provide **solid** evidence on how tree diversity affects host-parasitoid communities of insects in forests. The work will be of interest to ecologists working on biodiversity conservation, community ecology, and food webs.

**Abstract** Environmental factors can influence ecological networks, but these effects are poorly understood in the realm of the phylogeny of host-parasitoid interactions. Especially, we lack a comprehensive understanding of the ways that biotic factors, including plant species richness, overall community phylogenetic and functional composition of consumers, and abiotic factors such as microclimate, determine host-parasitoid network structure and host-parasitoid community dynamics. To address this, we leveraged a 5-year dataset of trap-nesting bees and wasps and their parasitoids collected in a highly controlled, large-scale subtropical tree biodiversity experiment. We

tested for effects of tree species richness, tree phylogenetic, and functional diversity, and species and phylogenetic composition on species and phylogenetic diversity of both host and parasitoid communities and the composition of their interaction networks. We show that multiple components of tree diversity and canopy cover impacted both, species and phylogenetic composition of hosts and parasitoids. Generally, phylogenetic associations between hosts and parasitoids reflected nonrandomly structured interactions between phylogenetic trees of hosts and parasitoids. Further, host-parasitoid network structure was influenced by tree species richness, tree phylogenetic diversity, and canopy cover. Our study indicates that the composition of higher trophic levels and corresponding interaction networks are determined by plant diversity and canopy cover, especially via trophic links in species-rich ecosystems.

## Introduction

Understanding the ecological consequences of biodiversity loss is an increasingly important task in ecology, given the ongoing biodiversity crisis (*Isbell et al., 2023*). Representing the interdependencies among organisms, ecological networks reflect whether and how species interact with each other across trophic levels, playing an indispensable role in assessing ecosystem stability and integrity (*de Ruiter et al., 1995*; *Harvey et al., 2017*). Changes in network structure of higher trophic levels usually coincide with variations in their diversity and community composition, which could be in turn affected by the changes in producers via trophic cascades (*Barnes et al., 2018*; *Gonzalez et al., 2020*). However, we still lack a generalizable framework for how these networks and especially phylogenetic interdependencies among species respond to biodiversity loss in ecosystems, such as changes in the tree diversity of forests (*Tylianakis et al., 2008*; *Grossman et al., 2018*). To better understand species coexistence and its role for biodiversity conservation, we must further study the mechanisms on dynamics of networks from multiple perspectives (*Tittensor et al., 2014*; *Brondizio et al., 2019*), e.g., top-down or bottom-up. Previous studies have shown asymmetric effects of top-down and bottom-up across trophic levels (*Vidal and Murphy, 2018*), shaping multitrophic communities together (*Hunter et al., 1992*). Moreover, the diversity and community composition of higher trophic levels could also be driven by microclimate, through, e.g., vegetation structure, and canopy cover (*Fornoff et al., 2021*; *Perlík et al., 2023*).

The host-parasitoid networks that unite bottom-up and top-down processes in many ecosystems are prone to strong alterations due to environmental change (*Tylianakis et al., 2006*; *Jeffs and Lewis, 2013*). Insect parasitoids attack and feed on and eventually kill their insect hosts (*Godfray and Godfray, 1994*). Parasitoids are thought to be particularly sensitive to environmental changes (*Hance et al., 2007*), because species in higher trophic levels usually have smaller population sizes and the fluctuations in their host populations may cascade up to impact the parasitoids. Therefore, insect host-parasitoid systems are ideal for studying the relationships between community-level changes and species interactions (*Jeffs and Lewis, 2013*). Previous studies mainly focused on the influence of abiotic factors, such as elevation and habitat structure, on host-parasitoid interactions (e.g., *Valladares et al., 2012*; *Maunsell et al., 2015*; *Grass et al., 2018*), and interaction structure (e.g., *Cagnolo et al., 2011*). However, the role of multiple components of plant diversity (i.e., taxonomic, functional, and phylogenetic diversity) in modifying host-parasitoid interaction networks remains poorly explored (but see *Staab et al., 2016*). Recent studies mainly focus on basic diversity associations between hosts and parasitoids (*Ebeling et al., 2012*; *Schuldt et al., 2019*; *Guo et al., 2021*). These studies have demonstrated both direct and indirect effects (i.e., one pathway and more pathways via other variables) of plant diversity on both host and parasitoid diversity, possibly via increased niche space and resource availability (*Guo et al., 2021*). Nevertheless, how these patterns propagate to their interaction networks is still unclear. Moreover, the effects of changing plant composition extend beyond sole plant species richness. Namely, the other diversity components (e.g., plant phylogenetic diversity) have been shown to better predict diversity-dependent bottom-up effects on host-parasitoid networks (e.g., *Staab et al., 2016*; *Staab et al., 2021*). It is especially important to take phylogenetic dependencies (e.g., phylogenetic diversity or phylogenetic congruence) within and between the trophic levels into account (*Webb et al., 2002*; *Emerson and Gillespie, 2008*). This makes it vital to account for multiple dimensions of biodiversity (e.g., taxonomic, phylogenetic, functional) and relevant trophic interactions (*Peralta et al., 2015*; *Volf et al., 2017*; *Wang et al.,*

*2020*). These components might jointly affect host-parasitoid networks in a system with high species diversity. Forests have garnered special attention lately, because they represent complex and large ecosystems susceptible to global change (*De Frenne et al., 2021*; *Popkin, 2021*). Understanding how multiple dimensions of biodiversity modulate the effects of tree diversity loss on the structure and interaction strength of host-parasitoid networks clearly requires further study (*Staab et al., 2016*; *Fornoff et al., 2019*).

Here, we use standardized trap nests for solitary cavity-nesting bees, wasps, and their parasitoids in a large-scale subtropical forest biodiversity experiment to test how multiple dimensions of tree diversity and community composition influence host-parasitoid network structure. A multifaceted approach is particularly important when considering that associations between trophic levels might be nonrandom and phylogenetically structured (*Volf et al., 2018*; *Wang et al., 2020*). We aimed to discern the primary components of the diversity and composition of tree communities that affect higher trophic level interactions via quantifying the strength and complexity of associations between hosts and parasitoids. We expected that (a) multiple tree community metrics, such as species, phylogenetic, and functional diversity, and species community composition can structure host and parasitoid community compositions, especially via phylogenetic processes (e.g., lineages of trophic levels diverge and evolve over time), as species interactions often show phylogenetic conservatism (e.g., *Pellissier et al., 2013*; *Peralta et al., 2015*). Further, we hypothesized that (b) host-parasitoid networks will be more complex and stable with increasing tree species richness due to increased number of potential links from higher richness of hosts and parasitoids, and (c) both community and interaction network changes can also be related to abiotic factors, such as microclimate driven by canopy cover, which might play a role in structuring Hymenoptera communities (*Haddad et al., 2011*; *Fornoff et al., 2021*). By better understanding the tree diversity impacts, the phylogenetic relationships, and the effects of abiotic factors, we can begin building a generalized framework for understanding host-parasitoid interactions in forest ecosystems.

## Results

Overall, 34,398 brood cells were collected from 13,267 tubes across 5 years of sampling (2015, 2016, 2018, 2019, and 2020). Six families of hosts and seventeen families of parasitoids were identified. Among them, we found 56 host species (12 bees and 44 wasps, mean abundance and richness are 400 and 45, respectively, for each plot) and 50 parasitoid species (38 Hymenoptera and 12 Diptera, mean abundance and richness are 14 and 9, respectively, for each plot). The full species list and their abundances are given in *Supplementary file 1a*. Overall, our sampling was adequate for analysis (especially for hosts), as confirmed by the sampling completeness evaluation (*Figure 1—figure supplement 1*).

### Community composition of hosts and parasitoids

Host species composition was significantly related to the species and phylogenetic composition of the trees and parasitoid communities (see nonmetric multidimensional scaling [NMDS] axis scores in *Figure 1*; i.e., preserving the rank order of pairwise dissimilarities between samples), as well as to canopy cover, tree phylogenetic mean pairwise distance (MPD), elevation, and eastness (sine-transformed radian values of aspect) (*Figure 1a*, *Table 1*, *Supplementary file 1b*). Parasitoid species composition was significantly associated with host phylogenetic diversity, tree functional diversity, tree MPD, eastness, and elevation, and was significantly related to tree species composition, host species composition, and canopy cover (*Figure 1b*, *Table 1*, *Supplementary file 1c*). Host phylogenetic composition was affected by tree species composition, tree MPD, tree functional diversity, canopy cover, eastness, elevation, and was especially affected by parasitoid species and phylogenetic composition (*Figure 1c*, *Table 1*, *Supplementary file 1d*). For parasitoid phylogenetic composition, significant relationships were found with tree species and phylogenetic composition, host species composition, tree functional diversity, canopy cover, elevation, and eastness (*Figure 1d*, *Table 1*, *Supplementary file 1e*). The PERMANOVA also highlighted the important role of canopy cover for host and parasitoid community (*Supplementary file 1f–i*). The Mantel test revealed a consistent pattern with the NMDS analysis, highlighting a pronounced relationship between the species

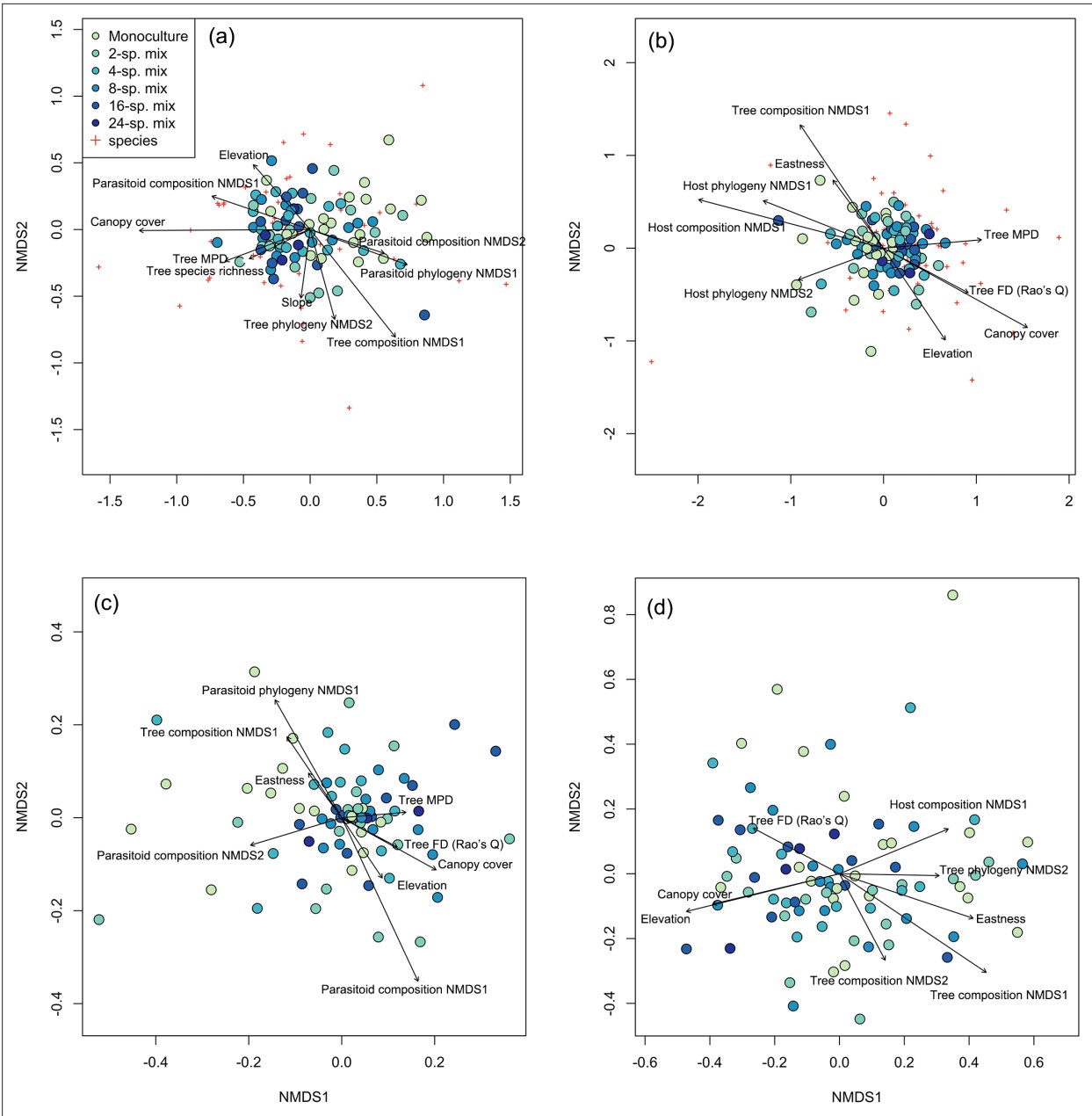

**Figure 1.** Associations among tree, host, and parasitoid species and phylogenetic composition. Ordination plot of the nonmetric multidimensional scaling (NMDS) analysis of (**a**) host species composition, (**b**) parasitoid species composition, (**c**) host phylogenetic composition, and (**d**) parasitoid phylogenetic composition across the study plots (filled circles) in the BEF-China experiment. Stress = 0.23, 0.23, 0.24, and 0.20, respectively. Arrows indicate significant (at p<0.05) correlations of environmental variables with NMDS axis scores. Lengths of arrows are proportional to the strength of the correlations. Red crosses refer to the host or parasitoid species in each community. See *Supplementary file 1b–e* in the Supplementary Materials for abbreviations and statistical values.

The online version of this article includes the following figure supplement(s) for figure 1:

**Figure supplement 1.** Sampling completeness assessment.

**Figure supplement 2.** Overview of the study plot distribution along the two experimental tree diversity sites of BEF-China (a: Site A, b: Site B).

composition of hosts and parasitoids (*Supplementary file 1j*). However, the correlation between the phylogenetic composition of hosts and parasitoids was not significant.

An effect of host composition on the composition of the parasitoid communities was further indicated by a significant parafit test (p=0.032) for testing the hypothesis of coevolution between a clade

**Table 1.** Environmental correlates of dissimilarity matrixes with predictors (nonmetric multidimensional scaling [NMDS] on Morisita-Horn dissimilarity) across the study plots.
Significant p-values are indicated in bold. See *Supplementary file 1b–e* for the complete information.

| | Host species community | Parasitoid species community | Host phylogenetic community | Parasitoid phylogenetic community |
|---|---|---|---|---|
| Tree phylogeny NMDS1 | 0.225 | 0.422 | 0.386 | 0.274 |
| Tree phylogeny NMDS2 | **0.003** | 0.12 | 0.128 | **0.024** |
| Tree composition NMDS1 | **0.001** | **0.001** | **0.001** | **0.001** |
| Tree composition NMDS2 | 0.604 | 0.418 | 0.433 | **0.031** |
| Canopy cover | **0.001** | **0.001** | **0.001** | **0.004** |
| Tree species richness | **0.035** | 0.122 | 0.100 | 0.094 |
| Elevation | **0.005** | **0.007** | **0.001** | **0.001** |
| Eastness | 0.079 | 0.045 | **0.04** | **0.001** |
| Northness | 0.49 | 0.837 | 0.821 | 0.340 |
| Slope | **0.031** | 0.507 | 0.507 | 0.959 |
| Tree FD (Rao's Q) | 0.094 | **0.019** | **0.021** | **0.031** |
| Tree MPD | **0.005** | **0.021** | **0.013** | 0.223 |
| Host phylogeny NMDS1 | – | **0.016** | – | 0.584 |
| Host phylogeny NMDS2 | – | **0.027** | – | 0.914 |
| Host composition NMDS1 | – | **0.001** | – | **0.008** |
| Host composition NMDS2 | – | 0.169 | – | 0.138 |
| Parasitoid phylogeny NMDS1 | **0.001** | – | **0.001** | – |
| Parasitoid phylogeny NMDS2 | 0.462 | – | 0.058 | – |
| Parasitoid composition NMDS1 | **0.001** | – | **0.001** | – |
| Parasitoid composition NMDS2 | **0.014** | – | **0.001** | – |

of hosts and a clade of parasites, suggesting nonrandom associations in the phylogenetic structure of parasitoid and host communities (*Figure 2*, *Figure 2—figure supplement 1*).

### Host-parasitoid network associations

The linear regression model results indicated that host vulnerability and linkage density were significantly positively associated with tree species richness. However, robustness of parasitoids was negatively correlated with tree species richness, while other environmental covariates had no significant effects (*Table 2*, *Figure 3*; except for elevation, which was marginally significantly related to robustness). Interaction evenness was significantly negatively associated with canopy cover, and interaction evenness was also negatively related to eastness (*Figure 4c*; *Table 2*). Parasitoid generality was only marginally associated with canopy cover, and was not related to tree species richness or the other environmental variables. In the alternative models (tree species richness replaced by tree MPD), host vulnerability and linkage density were significantly positively related to tree MPD (*Figure 4a and b*; *Supplementary file 1k*), while robustness of parasitoids was negatively related to tree MPD (*Figure 4—figure supplement 1*, *Supplementary file 1k*). The results of other network metrics (parasitoid generality and interaction evenness) were consistent with those of the primary models. Tree mean nearest taxon distance (MNTD) was unrelated to any network indices. It should be noted that the effects of tree species richness on host-parasitoid networks (i.e., host generality, parasitoid vulnerability, and linkage density) are more pronounced at one site. Since no directional differences were observed in the effects of tree species richness (*Table 2*), with all effects of tree species richness being

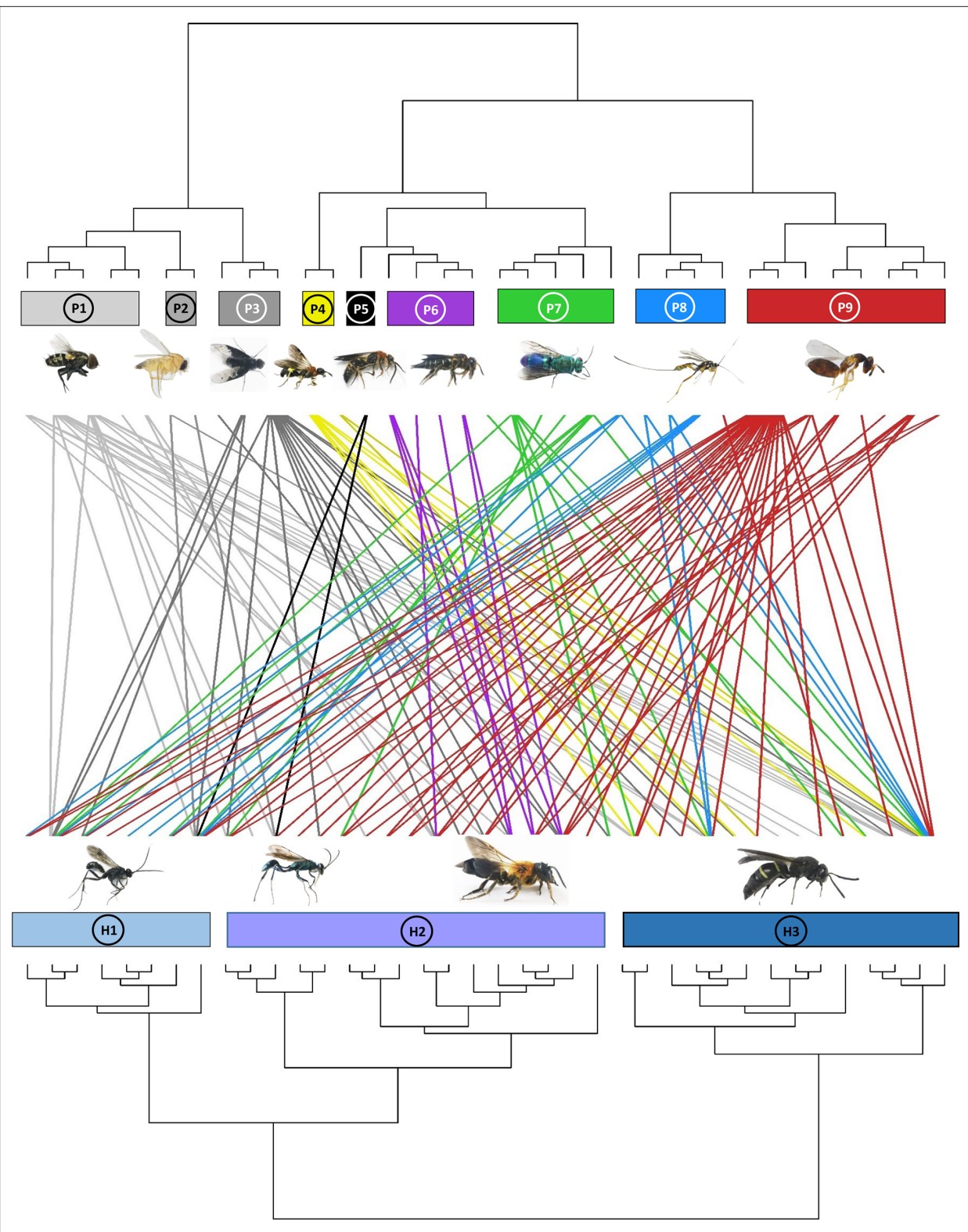

**Figure 2.** Dendrogram of phylogenetic congruence for the host species (below) and associated parasitoid species (above) recorded in the study. Each rectangle represents a different superfamily (for host species) or family (for parasitoid species). H1: Pompilidae, H2: Apoidea, H3: Vespidae; P1: Sarcophagidae, P2: Phoridae, P3: Bombyliidae, P4: Trigonalyidae, P5: Mutillidae, P6: Megachilidae, P7: Chrysididae, P8: Ichneumonidae, P9: Chalcidoidea. The trophic network of hosts and parasitoids was nonrandomly structured (parafit test: p=0.032). Host and parasitoid species names are given in *Figure 2—figure supplement 1*.

*Figure 2 continued on next page*

*Figure 2 continued*

The online version of this article includes the following figure supplement(s) for figure 2:

**Figure supplement 1.** Dendrogram of phylogenetic congruence for the host species (below) and associated parasitoid species (above) recorded in the study, showing the species name for hosts and parasitoids.

either positive or negative but one being significant, we have presented the overall pattern of these effects on the network indices (*Figures 3 and 4*).

The results of the null model analysis suggested that our metrics calculated by the observed network were significantly different from a random distribution (72, 71, and 77 out of 85 values for parasitoid generality, host vulnerability, and linkage density, respectively; all values for robustness, and interaction evenness), strongly demonstrating that interactions between species were not driven by random processes.

## Discussion

Our study demonstrates that tree species richness and phylogenetic diversity play key roles in modulating interacting communities of hosts and parasitoids. These interactions are further structured by the phylogenetic associations between hosts and parasitoids. Moreover, canopy cover partly determined host-parasitoid network patterns, including host vulnerability, linkage density, and interaction evenness. To some extent, this result supports a recent finding that the structure of host-parasitoid networks is also mediated by changes in microclimate (*Fornoff et al., 2021*), which is directly related to canopy cover. These patterns were highly associated with multiple tree diversity metrics (tree species, phylogenetic, and functional diversity), and compositional changes which are key to understand how host-parasitoid interactions may be impacted by biodiversity loss of lower trophic levels in food webs through trait- and phylogeny-based processes.

**Table 2.** Summary results of linear models for parasitoid generality, host vulnerability, robustness, linkage density, and interaction evenness of host-parasitoid network indices at the community level across the tree species richness gradient. Standardized parameter estimates (with standard errors, t- and p-values) are shown for the variables retained in the minimal models.

| | | Est. | SE | t | p |
|---|---|---|---|---|---|
| Parasitoid generality | Intercept | 0.176 | 0.016 | 10.96 | <0.001 |
| | Canopy cover | 0.033 | 0.016 | 2.03 | 0.046 |
| Host vulnerability | Intercept | 2.873 | 0.08 | 35.89 | <0.001 |
| | Elevation | –0.140 | 0.08 | –1.66 | 0.101 |
| | Tree species richness: Site A | 0.150 | 0.13 | 1.20 | 0.234 |
| | Tree species richness: Site B | 0.230 | 0.11 | 2.12 | 0.037 |
| Robustness of parasitoids | Intercept | 0.630 | 0.01 | 84.43 | <0.001 |
| | Tree species richness: Site A | –0.022 | 0.01 | –1.99 | 0.049 |
| | Tree species richness: Site B | –0.019 | 0.01 | –1.90 | 0.061 |
| | Intercept | 2.038 | 0.04 | 50.99 | <0.001 |
| | Elevation | –0.078 | 0.04 | –1.85 | 0.069 |
| | Tree species richness: Site A | 0.106 | 0.06 | 1.70 | 0.094 |
| | Tree species richness: Site B | 0.106 | 0.04 | 2.68 | 0.009 |
| Linkage density | Intercept | 0.511 | 0.009 | 59.12 | 0.025 |
| Interaction evenness | Canopy cover | –0.037 | 0.007 | –5.06 | <0.001 |
| | Eastness | –0.018 | 0.007 | –2.50 | 0.015 |

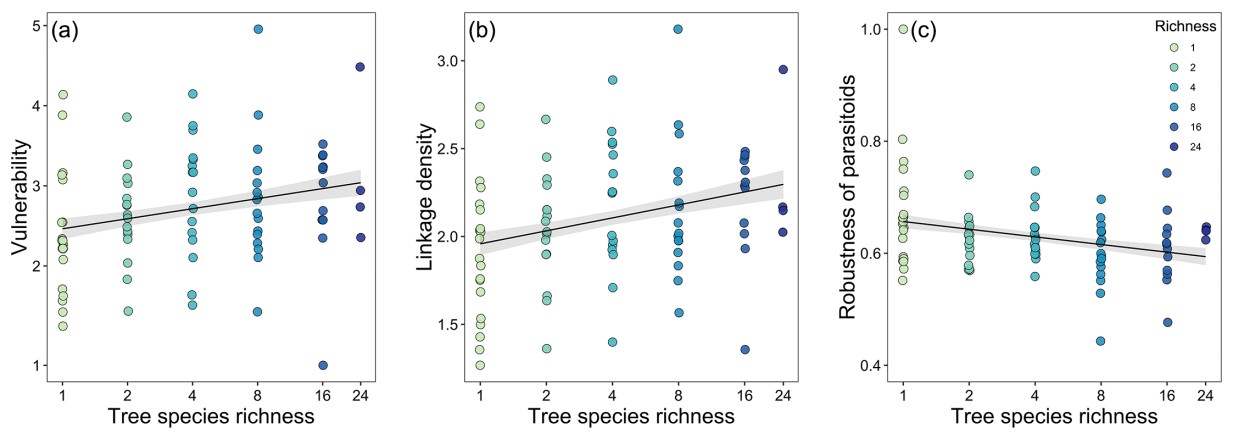

**Figure 3.** Bivariate relationships between tree species richness and network indices. Community-level relationships of network between tree species richness and (**a**) vulnerability, (**b**) linkage density, and (**c**) robustness of parasitoids. Values were adjusted for covariates of the final regression model. Regression lines (with 95% confidence bands) show significant (p<0.05) relationships. Note that axes are on a log scale for tree species richness.

The online version of this article includes the following figure supplement(s) for figure 3:

**Figure supplement 1.** Correlations among the predictors used in the study.

## Community composition associations

Host species composition was influenced by several factors, including the species and phylogenetic composition of trees and parasitoids, tree diversity (species richness and MPD), and other abiotic factors such as elevation and slope (*Figure 1a*; *Table 1*). The effects of tree diversity and composition on host species composition agree with previous study where solitary bee and wasp species compositions were related to plant community structure (e.g., *Loyola and Martins, 2008*). It seems likely that these results are based on bee linkages to pollen resources and predatory wasp linkages to the diverse of food sources, which may themselves be closely linked to resource heterogeneity increasing with tree species richness (*Reitalu et al., 2019*; *Staab and Schuldt, 2020*).

We also found that tree MPD, FD, and species composition affect parasitoid species composition (*Figure 1b*; *Table 1*), similarly as other studies that have found significant relationships between plant

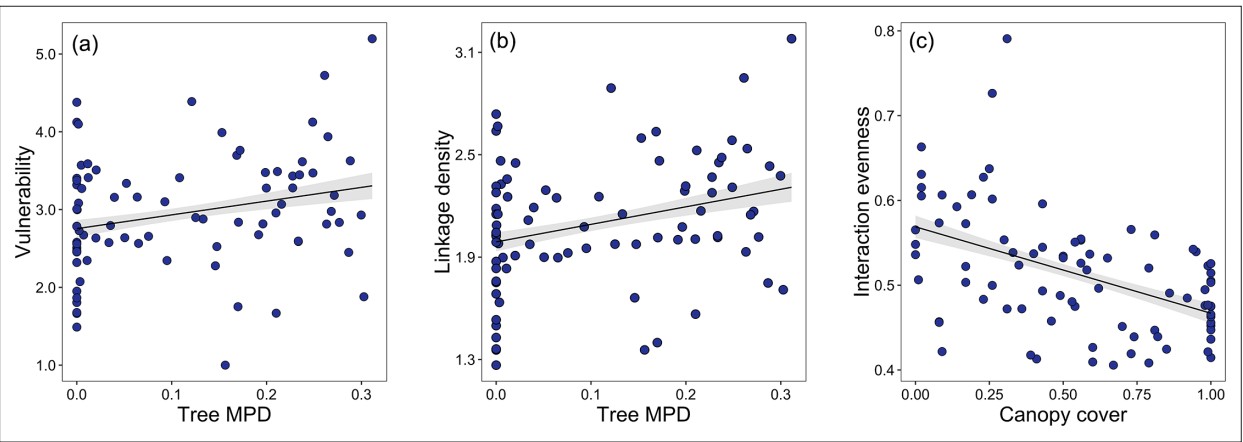

**Figure 4.** Bivariate relationships between tree MPD, canopy cover and network indices. Community-level relationships of network between tree phylogenetic mean pairwise distance and (**a**) vulnerability and (**b**) linkage density and community-level relationships of network between canopy cover, and (**c**) interaction evenness. Values were adjusted for covariates of the final regression model. Regression lines (with 95% confidence bands) show significant (p<0.05) relationships.

The online version of this article includes the following figure supplement(s) for figure 4:

**Figure supplement 1.** Community-level relationships of networks between tree phylogenetic mean pairwise distance (MPD) and robustness of parasitoids.

and parasitoid diversity in forests, including tree phylogenetic diversity (*Staab et al., 2016*), functional diversity (*Rodríguez et al., 2019*), and structural diversity (*Schuldt et al., 2019*). Similar to predators (*Chen et al., 2023*), parasitoids might also be more active or efficient with increasing tree community dissimilarity due to higher prey resources or lower intraguild parasitism caused by more diverse habitats (*Finke and Denno, 2002*). On the other hand, our results also show that host species composition and parasitoid species composition relate to each other and their phylogenetic compositions, which are structured by tree communities to some extent. This pattern could propagate to the adjacent two higher trophic interactions through both top-down and bottom-up control.

Both host and parasitoid phylogenetic compositions were related to tree species composition (*Figure 1c and d*; *Table 1*). This pattern has important implications for cascading effects among trophic levels, in that producer communities (i.e., trees) could structure a higher trophic level community (i.e., parasitoids) via an intermediate trophic level (i.e., hosts). However, only parasitoids responded to tree phylogenetic composition. This may be because there are many caterpillar-hunting wasps in our host communities, and the community composition of caterpillars were usually correlated with tree phylogenetic communities (*Wang et al., 2019*). Therefore, the prey organisms highly associated with tree phylogenetic composition (e.g., caterpillars) might indirectly determine predatory wasp (host) phylogenetic composition, similarly as recently found for interactions between plants, caterpillars, and spiders (*Chen et al., 2023*). This could be further tested by analyzing the availability of food directly used by the wasps (e.g., caterpillars). For parasitoids, tree phylogenetic composition might drive the process of community assembly through trophic cascades (e.g., from plants to parasitoids via herbivores and host wasps) (*Webb et al., 2002*; *Cavender-Bares et al., 2009*). Additionally, parasitoid phylogenetic composition can be indirectly influenced by tree canopy structural diversity (e.g., host availability in plots with higher heterogeneity; *Schuldt et al., 2019*), which can be determined by conserved traits across tree phylogenies (*Webb et al., 2002*). The phylogenetic associations between hosts and parasitoids exhibited a nonrandom structure (significant *parafit* correlation; *Figure 2*) between the phylogenetic trees of the host and their parasitoids (see also *Peralta et al., 2015*). Such pattern could be further confirmed by the significant association between host phylogenetic composition and parasitoid phylogenetic composition (*Figure 1c*), which suggested that their interactions are phylogenetically structured to some extent. However, this significant pattern was observed only in the NMDS analysis and not in the Mantel test, suggesting that the nonrandom interactions between hosts and parasitoids cannot be solely explained by community similarity. Instead, the phylogenetic associations between hosts and parasitoids appear to be more intricate, warranting further investigation in future studies.

Moreover, the species and phylogenetic compositions of hosts and parasitoids were also related to canopy cover, which has been considered especially important to microclimate (*Sobek et al., 2009*; *Fornoff et al., 2021*). In future studies, it will be useful to incorporate other, more direct metrics of microclimate, such as local temperature and humidity, to determine the proximal drivers of these microclimatic effects (*Ma et al., 2010*; *Fornoff et al., 2021*).

## Community-level host-parasitoid networks

Tree species richness did not significantly influence the diversity of hosts targeted by parasitoids (parasitoid generality), but caused a significant increase in the diversity of parasitoids per host species (host vulnerability) (*Figure 3a*; *Table 2*). This is likely because niche differentiation often influences network specialization via potential higher resource diversity in plots with higher tree diversity (*López-Carretero et al., 2014*). Such positive relationship between host vulnerability and tree species richness suggested that host-parasitoid interactions could be driven through bottom-up effects via benefit from tree diversity. For example, parasitoid species diversity increases more than host diversity with increasing tree species richness (*Guo et al., 2021*), resulting in an increase of host vulnerability at community level. According to the enemies hypothesis (*Root, 1973*), which predicts a positive effects of increasing plant richness on natural enemies of herbivores, the higher trophic levels in our study (e.g., predators and parasitoids) would benefit from tree diversity and, thereby, regulate herbivores (*Staab and Schuldt, 2020*). Indeed, previous studies at the same site found that bee parasitoid richness and abundance were positively related to tree species richness, but not to the abundance and richness of their bee hosts (*Fornoff et al., 2021*; *Guo et al., 2021*). Because our dataset considered all hosts and reflects an overall pattern of host-parasitoid interactions, the effects of tree species

richness on parasitoid generality might be more complex and difficult to predict, as we found that neither tree species richness nor tree MPD were related to parasitoid generality. Future research should further explore these patterns by incorporating temporal scales and decomposing interactions across different functional groups (e.g., pollinators such as bees and predators such as wasps).

Linkage density was positively related to tree species richness (*Figure 3b*; *Table 2*), supporting the food web theory, which predicts in our case that network complexity (linkage density) depends on the number of plants (*Blüthgen and Staab, 2024*). It was expected that higher trophic levels would be more robust, less influenced by perturbations from lower trophic levels, when plant diversity is higher, as more potential interactions at lower trophic levels should theoretically increase redundancy and resilience of connected higher levels (*Blüthgen and Klein, 2011*; *Fornoff et al., 2019*). Although trees were not directly included as a trophic level in our networks, potential network complexity increased with tree species richness, likely enabling higher network stability/resistance (*Ebeling et al., 2011*; *Staab et al., 2015*). For example, a network might be more sensitive to extinctions because of key species loss due to lower linkage density and lower redundancy (*Blüthgen and Staab, 2024*). However, parasitoid robustness was negatively related to tree species richness (*Figure 3c*; *Table 2*). Dilution effects may explain this, as plots with higher richness held fewer individuals of a given tree species. If there are strong prey item (caterpillars, grasshoppers, etc.) preferences for one species, there may be fewer individuals per area or they may be more densely aggregated and less likely to be encountered by parasitoids. This increased stochasticity in parasitoid wasps could benefit hosts by reducing parasitism pressure overall, weakening top-down controls.

Similar to tree species richness, tree MPD was also positively correlated with host vulnerability and linkage density (*Figure 4a and b*), meaning that the mean number of parasitoids per host species and number of links within the host-parasitoid system can also be promoted by tree MPD. This is in agreement with several recent studies showing that plant phylogenetic diversity not only affects herbivores but also higher tropic levels (*Pellissier et al., 2013*; *Staab and Schuldt, 2020*; *Wang et al., 2020*). Our results suggest that the specialization and complexity of higher trophic levels can also be affected by plant phylogenetic diversity. This pattern can be traced to the effects of habitat heterogeneity caused by tree species richness and MPD on higher trophic levels via bottom-up control. The effects of tree MPD were consistent with effects of tree species richness on robustness of parasitoids to host loss. This result suggests that higher trophic levels are sensitive to changes in both plant phylogenetic relatedness and taxonomic species dissimilarity via trophic interactions, even the hosts are not all directly interacting with plants, bees excluded. Therefore, it may be that stronger linkages (e.g., more pronounced diversity effects on the network structure) would be found when exclusively exploring such plant-herbivore-parasitoid systems.

Interaction evenness was significantly negatively related to canopy cover (*Figure 4c*), further reinforcing the important role of canopy cover-modulated microclimate (likely temperature and humidity) for trophic interactions (*Sobek et al., 2009*; *Fornoff et al., 2021*). Our results agree with a previous study on ants, where plant-insect interactions were more even with more open canopies (*Dáttilo and Dyer, 2014*). In our case, canopy cover might change Hymenoptera species evenness and then further influence interaction evenness. Certain host species tended to nest in plots with higher canopy cover, which might decrease the interaction evenness by favoring parasitoids of fewer, more dominant hosts. This pattern would become more significant when more host and parasitoid species are in a plot, given the positive relationship between higher trophic level diversity and canopy cover.

## Future prospects

Overall, our study enables new insights into the dynamics of host-parasitoid interactions under varying canopy conditions, an important step toward building a synthetic model for such biodiversity. A key finding was that although parasitoids and hosts respond to tree species richness, the effects of parasitoids on hosts were more pronounced than those of hosts on parasitoids. However, whether this pattern holds for other antagonistic interactions requires further investigation. Different trophic levels and functional groups of species responded differently to experimental changes in plant communities (*Fornoff et al., 2021*; *Guo et al., 2021*). This highlights the complexities of building multitrophic networks and calls for more studies across habitat types and taxa, to test the generality of our findings. Future studies should also consider the role of host/parasitoid functional traits, because they might play a critical role in modifying network structures and ecosystem functioning.

## Materials and methods

### Study sites design

This study was conducted in the BEF-China biodiversity experiment, which is the largest tree diversity experiment worldwide. The experiment is located in a subtropical forest near Xingangshan, Jiangxi province, south-east China (29°08′–29°11′N, 117°90′–117°93′E). The mean annual temperature is 16.7°C and mean annual precipitation 1821 mm (*Yang et al., 2013*). The experiment includes two study sites (Site A and Site B), 4 km apart from each other, that were established in 2009 (Site A) and 2010 (Site B), respectively. A total of 566 plots (25.8 m×25.8 m) were designed on the two sites, and per plot 400 trees were initially planted in 20 rows and 20 columns with a planting distance of 1.29 m. A tree species richness gradient (1, 2, 4, 8, 16, and 24 species) was established at each site, based on a species pool of 40 local, broadleaved tree species (*Bruelheide et al., 2014*). The tree species pools of the two plots are nonoverlapping (16 species for each site). The composition of tree species within the study plots is based on a 'broken-stick' design (see *Bruelheide et al., 2014*).

For our study, at both sites (Site A and Site B) eight plots of each tree species richness level (1, 2, 4, 8) were randomly selected, as well as six and two plots of 16 and 24 mixtures. In addition, at Site B eight additional monocultures were sampled (*Fornoff et al., 2021*), resulting in 48 plots in Site B, including 16 monocultures, eight plots for each 2, 4, 8 mixtures and six and two plots for 16 and 24 mixtures (there are 8 overlapping tree species across 24 mixtures at the two sites). In total, 88 study plots were used (40 plots on Site A and 48 plots on Site B, see *Figure 1—figure supplement 2*).

### Sampling

We collected trap nests monthly to sample solitary bees and wasps (*Staab et al., 2018*) in the 88 plots from September to November in 2015 and April to November in 2016, 2018, 2019, and 2020. For each plot, we installed two poles with trap nests (11 m apart from each other and 9 m away from the nearest adjacent plots) along a SW-NE diagonal (following the design of *Ebeling et al., 2012*). Each pole stood 1.5 m above ground, and each trap nest consisted of two PVC tubes (length: 22 cm ×diameter: 12.5 cm) filled with 75±9 (SD) reed internodes of 20 cm length and diameters varying between 0.1 and 2.0 cm (*Staab et al., 2014*; *Fornoff et al., 2021*). Every month, we sampled the reeds with nesting hymenopterans and replaced them with internodes of the same diameter. All the samples were reared in glass test tubes under ambient conditions until specimens hatched. We identified hatched hosts and parasitoids to species or morphospecies (*Supplementary file 1a*) based on reference specimens (vouchered at the Institute of Zoology, CAS, Beijing). We were interested in the general patterns of host-parasitoid interactions at the community level, so for the analysis we did not distinguish between the two life-history strategies of parasitoids (true parasitoids and kleptoparasitoids, including hymenopteran and dipteran parasitoids) because they both have the same ecological result, death of host brood cells. We evaluated our sampling completeness with r package '*iNEXT*' (*Hsieh et al., 2016*).

### DNA extraction and amplification

All specimens were sequenced for a region of the mitochondrial cytochrome *c* oxidase subunit I (COI) gene (*Hebert et al., 2003*). We extracted whole-genomic DNA of hosts and parasitoids using DNeasy Blood & Tissue Kits (QIAGEN GmbH, Hilden, Germany), following the manufacturer's protocols. COI sequences of samples were amplified using universal primer pairs, LCO1490 (GGTCAACAAATCATAAAGATATTGG) as the forward primer and HCO2198 (TAAACTTCAGGG TGACCAAAAAATCA) or HCOout (CCAGGTAAAATTAAAATATAAACTTC) as the reverse primer. We carried out polymerase chain reactions (PCRs) in 96-well plates with 30 µl reactions containing 10 µl ddH$_2$O, 15 µl Premix PrimeSTAR HS (TaKaRa), 1 µl of each primer at 10 µM, and 3 µl template genomic DNA using a thermo cycling profile. The PCR procedure is as follows: 94°C for 1 min; 94°C for 1 min, 45°C for 1.5 min and 72°C for 1.5 min, cycle for 5 times; 94°C for 2 min, 58°C for 1.5 min and 72°C for 1 min, cycle for 36 times; 72°C for 5 min. We performed all PCRs on an Eppendorf Mastercycler gradient, which were then visualized on a 1% agarose gel. Samples with clean single bands were sequenced after PCR purification using BigDye v3.1 on an ABI 3730xl DNA Analyser (Applied Biosystems).

## Sequence alignment and phylogenetic analysis

We applied MAFFT (*Katoh et al., 2002*) to align all sequences, then translated the nucleotides into amino acids via MEGA v7.0 (*Kumar et al., 2016*) to check for the presence of stop codons with manual adjustments. Host and parasitoid sequences were then aligned against the references using a Perl-based DNA barcode aligner (*Chesters, 2019*).

We employed two strategies to improve the phylogenetic structure of a DNA barcode phylogeny, which demonstrably improve resulting phylogeny-based diversity indices (*Macías-Hernández et al., 2020*). These include the integration of (a) molecular sequences of the plot data and (b) phylogenetic relationships from other molecular datasets. Integration was achieved following *Wang et al., 2020*, and *Chesters, 2020*: reference DNA barcodes of Hymenoptera and Diptera were downloaded from the BOLD API (https://bench.boldsystems.org/index.php/API_Public), which were variously processed (e.g., to retain only fully taxonomically labeled barcodes, to remove low-quality or mislabeled entries, and to dereplicate to a single exemplar per species), and then aligned (*Chesters, 2019*). A single outgroup was included for which we selected the most appropriate insect order sister to Diptera and Hymenoptera (*Misof et al., 2014*), a representative of the order Psocoptera (Psocidae, *Psocus leidyi*). We then constructed a phylogeny of the references and subjects, with references constrained according to the method described earlier (*Chesters, 2020*). A number of backbone topologies were integrated for setting hard and soft constraints, including a transcriptomics-derived topology (*Chesters, 2020*), a mitogenome tree of insects (*Chesters, 2017*), Diptera-specific trees (*Wiegmann et al., 2011*; *Cranston et al., 2012*; *Ament, 2017*), and Hymenoptera-specific trees (*Peters et al., 2011*; *Branstetter et al., 2017*; *Cardinal, 2018*). The constrained inference was conducted with RaxML version 8 (*Stamatakis, 2014*) under the standard GTRGAMMA DNA model with 24 rate categories. According to the backbone trees used, most taxa present were monophyletic with a notable exception of Crabronidae, for which there is emerging phylogenomic evidence of its polyphyly (*Sann et al., 2018*).

## Tree phylogenetic diversity, functional diversity, and environmental covariates

The phylogenetic diversity of the tree communities was quantified by calculating wood volume-weighted phylogenetic MPD (*Tucker et al., 2017*). Tree wood volume was estimated from data on basal area and tree height (*Bongers et al., 2021*) measured in the center of each plot. Moreover, to represent variations toward the tips of the phylogeny beyond MPD, we additionally calculated MNTD, which is a measure that quantifies the distance between each species and its nearest neighbor on the phylogenetic tree (*Webb, 2000*). Phylogenetic metrics of trees (tree MPD and MNTD) were calculated based on a maximum likelihood phylogenetic tree available for the tree species in our study area (*Michalski et al., 2017*). Considering that predatory wasps mainly feed on herbivorous caterpillars, we calculated tree functional diversity to test the indirect effects on hymenopteran communities and relevant network indices. Specifically, seven leaf traits were used for calculation of tree functional diversity, which was calculated as the MPD in trait values among tree species, weighted by tree wood volume, and expressed as Rao's Q (*Ricotta and Moretti, 2011*), including specific leaf area, leaf toughness, leaf dry matter content, leaf carbon content, ratio of leaf carbon to nitrogen, leaf magnesium content, and leaf calcium content. These functional traits were commonly related to higher trophic levels in our study area, such as herbivores and predators (*Wang et al., 2020*; *Chen et al., 2023*), which are the main food resources of our predatory wasps. All of the traits were measured on pooled samples of sun-exposed leaves of a minimum of five tree individuals per species following standard protocols (*Pérez-Harguindeguy et al., 2003*).

As our analyses mainly compare community patterns among study plots, we additionally considered potential effects of environmental variation by using plot means of slope, elevation, 'eastness' (sine-transformed radian values of aspect), and 'northness' (cosine-transformed radian values of aspect) as environmental covariates that characterize the heterogeneity of the study plots. We also accounted for the potential effects of canopy cover at plot level for host-parasitoid interactions, as it can structure hymenopteran communities (*Perlík et al., 2023*). Canopy cover was calculated as in *Fornoff et al., 2021*, based on hemispherical photographs.

## Statistical analysis

All analyses were conducted in R 4.1.2 with the packages *ape*, *vegan*, *picante*, *bipartite*, and *caper* (http://www.R-project.org). Prior to analysis, samples from the 5 years (2015, 2016, 2018, 2019, and 2020) were pooled at the plot level to discern overall and generalizable effects permeating this system. We excluded three plots with no living trees because of high mortality, resulting in 85 plots in the final analysis.

## Composition of trees, hosts, and parasitoids

The species and phylogenetic composition of trees, hosts, and parasitoids were quantified with NMDS analysis based on Morisita-Horn distances. The minimum number of required dimensions in the NMDS based on the reduction in stress value was determined in the analysis (*k*=2 in our case). We centered the results to acquire maximum variance on the first dimension, and used the principal components rotation in the analysis. The phylogenetic composition was calculated by MPD among the host or parasitoid communities per plot with the R package '*picante*' applying the '*mpd*' function. To test the influence of study plot heterogeneity on these relationships, we fitted their standardized values (see vectors in *Supplementary file 1b*) to the ordination on the basis of a regression with the NMDS axis scores (*Quinn and Keough, 2002*). NMDS was widely used to summarize the variation in species composition across plots. The two axes extracted from the NMDS represent gradients in community composition, where each axis reflects a subset of the compositional variation. These axes should not be interpreted in isolation, as the overall species composition is co-determined by their combined variation. For clarity, results were interpreted based on the relationships of variables with the compositional gradients captured by both axes together. For the analysis, we considered tree species richness, tree functional and phylogenetic diversity, canopy cover, and environmental covariates (elevation, eastness, northness, and slope) as plot characteristics. We assessed the significance of correlations with permutation tests (permutation: *n*=999). To strengthen the robustness of our findings based on NMDS, we further validated the composition results using Mantel test and PERMANOVA (with 'adonis2') for correlation between communities and relationships between communities and environmental variables.

## Phylogenetic match of hosts and parasitoids

In addition, we used a parafit test (9999 permutations) with the R package '*ape*' to test whether the associations were nonrandom between hosts and parasitoids. This is widely used to assess host-parasite co-phylogeny by analyzing the congruence between host and parasite phylogenies using a distance-based matrix approach. The species that were not attacked by parasitoids or failed to generate sequences were excluded from the analyses. For species abundance and composition, see *Supplementary file 1a*.

## Host-parasitoid interactions

We constructed quantitative host-parasitoid networks at community level with the R package '*bipartite*' for each plot of the two sites. Bees and wasps were considered together as hosts because there were too few abundant bee species to analyze separate interaction networks for bees and wasps. We calculated five indices to quantitatively characterize the structure of the interaction networks (*Blüthgen and Staab, 2024*): weighted parasitoid generality (effective number of host species per parasitoid species), weighted host vulnerability (effective number of parasitoid species attacking a host species), robustness (degree of network stability), linkage density (degree of network specialization), and interaction evenness (degree of network evenness). Parasitoid generality was defined as the weighted mean number of host species per parasitoid species, $G_{qw} = \sum_{j=1}^{J} \frac{A_j}{m} 2^{H_j}$, with $A_j$ being the number of interactions of parasitoid species *j*, *m* the total number of interactions of all species, and $H_j$ the Shannon diversity of interactions of parasitoid species *j*. Host vulnerability was the weighted mean number of parasitoid species per host species, vulnerability $= \sum_{i=1}^{I} \frac{A_i}{m} 2^{H_i}$ (*Bersier et al., 2002*). Robustness was defined as the area under the extinction curve, reflecting the degree of decreases of one trophic level with the random elimination species of the other trophic levels, here using the robustness index for higher trophic levels (i.e., parasitoids). For linkage density, $L_q = 0.5 \left( \sum_{j=1}^{J} \frac{A_j}{m} 2^{H_j} + \sum_{i=1}^{I} \frac{A_i}{m} 2^{H_i} \right)$, we used the realized proportion of possible links between the two trophic levels as the mean number of

interactions per species across the entire network (*Tylianakis et al., 2007*). Interaction evenness was defined as $E_s = -\sum_i \sum_j p_{ij} \ln p_{ij} / \ln IJ$, which is used to describe Shannon's evenness of network interactions (*Dormann et al., 2009*). To check whether all network indices significantly differ from chance across all study plots, we used Patefield null models (*Dormann et al., 2009*) to compare observed indices with simulated values (10,000 times).

## Linear models

To test the effects of tree species richness, tree phylogenetic, and functional diversity, as well as canopy cover and the other environmental covariates (including slope, elevation, eastness, and northness) on the five network indices (vulnerability, generality, linkage density, interaction evenness, robustness of parasitoids), we used simple linear models. For our analyses, we included the interactions between site and tree species richness, site and tree MPD, and site and tree functional diversity as predictors. Given the strong correlation between tree species richness and tree MPD (Pearson's r=0.74, p<0.01), we excluded tree MPD in the models where tree species richness was a predictor. To evaluate the potential effects caused by tree MPD, we also ran alternative models where tree species richness was replaced with tree MPD. We simplified all models by gradually removing nonsignificant factors to obtain the most parsimonious model with the lowest AICc (*Table 2*, *Supplementary file 1k*). To ensure that the analyses were not strongly affected by multicollinearity, the correlations among all predictors were tested (*Figure 3—figure supplement 1*), and variance inflation factors of our statistical models were checked.

## Acknowledgements

We thank the BEF-China consortium for support (especially, Bo Yang). We thank Mr. Yinquan Qi for his help with the collection. This work was supported by the National Key Research Development Program of China (2022YFF0802300), the National Science Fund for Excellent Young Scholars (32122016), the Strategic Priority Research Program of the Chinese Academy of Sciences (XDB310304), the National Natural Science Foundation, China (32100343, 32070465), and the National Science Fund for Distinguished Young Scholars (31625024). MQW was supported by the Alexander von Humboldt research fellowships. CDZ's lab is funded by the Key Program of the National Natural Science Foundation of China (No. 32330013) and also has been continuously supported by grants from the Key Laboratory of the Zoological Systematics and Evolution of the Chinese Academy of Sciences (grant number 2008DP173354).

## Additional information

### Funding

| Funder | Grant reference number | Author |
|---|---|---|
| The National Key Development program of China | 2022YFF0802300 | Arong Luo |
| The National Natural Science Foundation, China | 32100343 | Ming-Qiang Wang |
| The National Science Fund for Excellent Young Scholars | 32122016 | Arong Luo |
| The Strategic Priority Research Program of the Chinese Academy of Sciences | XDB310304 | Chao-Dong Zhu |
| The National Natural Science Foundation, China | 32070465 | Arong Luo |

| Funder | Grant reference number | Author |
|---|---|---|
| The Alexander von Humboldt research fellowships | | Ming-Qiang Wang |
| The Key Program of the National Natural Science Foundation of China | 32330013 | Chao-Dong Zhu |
| The Key Laboratory of the Zoological Systematics and Evolution of the Chinese Academy of Science | 2008DP173354 | Chao-Dong Zhu |
| The National Science Fund for Distinguished Young Scholars | 31625024 | Chao-Dong Zhu |

The funders had no role in study design, data collection and interpretation, or the decision to submit the work for publication.

## Author contributions

Ming-Qiang Wang, Conceptualization, Formal analysis, Validation, Methodology, Writing – original draft, Writing – review and editing; Shi-Kun Guo, Data curation, Methodology, Writing – review and editing; Peng-Fei Guo, Data curation, Investigation, Writing – review and editing; Juan-Juan Yang, Guo-Ai Chen, Data curation, Investigation; Douglas Chesters, Methodology, Writing – original draft; Michael C Orr, Writing – review and editing; Ze-Qing Niu, Jing-Ting Chen, Yi Li, Qing-Song Zhou, Felix Fornoff, Xiaoyu Shi, Shan Li, Methodology; Michael Staab, Massimo Martini, Andreas Schuldt, Methodology, Writing – review and editing; Alexandra-Maria Klein, Methodology, Project administration, Writing – review and editing; Xiaojuan Liu, Keping Ma, Helge Bruelheide, Methodology, Project administration; Arong Luo, Supervision, Funding acquisition, Validation, Methodology, Project administration; Chao-Dong Zhu, Resources, Supervision, Funding acquisition, Methodology, Project administration, Writing – review and editing

## Author ORCIDs

Ming-Qiang Wang https://orcid.org/0000-0002-3175-2200
Michael C Orr https://orcid.org/0000-0002-9096-3008
Felix Fornoff https://orcid.org/0000-0003-0446-7153
Massimo Martini https://orcid.org/0000-0002-1855-9334
Alexandra-Maria Klein https://orcid.org/0000-0003-2139-8575
Xiaojuan Liu https://orcid.org/0000-0002-9292-4432
Arong Luo https://orcid.org/0000-0001-9652-5896
Chao-Dong Zhu https://orcid.org/0000-0002-9347-3178

Reviewer #2 (Public review): https://doi.org/10.7554/eLife.100202.3.sa1
Author response https://doi.org/10.7554/eLife.100202.3.sa2

# Additional files

## Supplementary files

Supplementary file 1. Supplementary tables were included in this file.

MDAR checklist

## Data availability

Data is available on Science Data Bank at https://doi.org/10.57760/sciencedb.09199 and on the BEF-China project database at https://data.botanik.uni-halle.de/bef-china/datasets/684.

The following datasets were generated:

| Author(s) | Year | Dataset title | Dataset URL | Database and Identifier |
|---|---|---|---|---|
| Wang MQ, Luo A, Zhu CD | 2024 | Network indices of trap nest in BEF-China | https://doi.org/10.57760/sciencedb.09199 | Science Data Bank, 10.57760/sciencedb.09199 |
| Wang et al. | 2024 | Wang eLife 2024 trap nest data | https://data.botanik.uni-halle.de/bef-china/datasets/684 | BEF-China, 684 |

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
