## [Editor Report · eLife Assessment]

This **valuable** study uses a massive and long-term experimental data set to provide **solid** evidence on how tree diversity affects host-parasitoid communities of insects in forests. The work will be of interest to ecologists working on biodiversity conservation, community ecology, and food webs.

---

## [Referee Report · Reviewer #2 (Public review)]

Summary

The authors use a tree biodiversity experiment to evaluate the effects of tree community and canopy cover on communities of cavity-nesting Hymenoptera and their parasitoids and the interactions between these two guilds. They find that multiple measures of tree diversity influence the hosts, parasitoids, and their interactions. In addition, host-parasitoid interactions show a phylogenetic signal.

Strength

The authors use a massive, long-term data set, meaningful community descriptors, and a solid set of analyses to explore the impacts of tree communities on host-parasitoid networks. It is rare to have such detailed data from multiple different trophic levels.

Weakness

Even though the data expands over several seasons, this is not considered in the analyses, but communities sampled at different years are pooled at the plot level. A more detailed analysis of the variations between years could reveal underlaying patterns as currently the differences in the communities and their structure between the years are ignored (e.g., when estimating the phylogenetic compositions not all the species pooled together actually coexist in time).

Also, the precision of the writing should be improved as it was not always easy to follow the text and the thoughts.

---

## [Author Response]

The following is the authors’ response to the original reviews.

It would be great if the authors could add clarification about the NMDS analyses and the associated results (Fig. 1, Table 1 and Tables S2-4). The overall aim of these analyses was to see how plot characteristics (e.g. canopy cover) and composition of one taxonomic group were related to the composition of another taxonomic group. The authors quantified species composition by two axes from NMDS. (1) This analysis may yield an interpretation problem: if we only find one of the axes, but not the other, was significantly related to one variable, it would be difficult to determine whether that specific variable is important to the species composition because the composition is co-determined by two axes. (2) It is unclear how the authors did the correlation analyses for Tables S2-4. If correlation coefficients were presented in these tables, then these coefficients should be the same or very similar if we switch the positions of y vs. x. That is, the correlation between host vs. parasite phylogenetic composition would be very close to the correlation between parasite vs. phylogenetic composition, but not as the author found that these two relationships were quite different, leading to the interpretation of bottom-up or top-down processes. It is also unclear which correlation coefficient was significant or not because only one P value was provided per row in these tables. (3) In addition to the issues of multiple axes (point 1), NMDS axes simply define the relative positions of the objects in multi-dimensional space, but not the actual dissimilarities. Other methods, such as generalized dissimilarity modeling, redundancy analysis and MANOVA, can be better alternatives.

Thank you for the thorough and constructive review. We have taken the concerns and questions raised by the editors and reviewers into account and provided clarification about the NMDS analyses as well as additional analyses to confirm our results. First, we have now added a brief explanation in the manuscript regarding the interpretation of the two NMDS axes and how they relate to species composition. Specifically, we clarified that while NMDS defines the relative positions of objects in multi-dimensional space, the two axes together provide a more comprehensive representation of the community composition, which is not solely determined by either axis independently. Second, we acknowledge that alternative approaches could help further strengthen our conclusions. To address this, we incorporated Mantel tests and PERMANOVA (with ‘adonis2’) as additional validation methods. These analyses allowed us to summarize compositional patterns while testing our hypotheses within the framework of the plot characteristics and taxonomic relationships. We have added these analyses and their results in the manuscript to reinforce our findings.

In methods: L478-481 “To strengthen the robustness of our findings based on NMDS, we further validated the results using Mantel test and PERMANOVA (with ‘adonis2’) for correlation between communities and relationships between communities and environmental variables.”

L469-475 “NMDS was used to summarize the variation in species composition across plots. The two axes extracted from the NMDS represent gradients in community composition, where each axis reflects a subset of the compositional variation. These axes should not be interpreted in isolation, as the overall species composition is co-determined by their combined variation. For clarity, results were interpreted based on the relationships of variables with the compositional gradients captured by both axes together."

In results: L172-177 “The PERMANOVA analysis also highlighted the important role of canopy cover for host and parasitoid community (Table S6-9). The Mantel test revealed a consistent pattern with the NMDS analysis, highlighting a pronounced relationship between the species composition of hosts and parasitoids (Table S10). However, the correlation between the phylogenetic composition of hosts and parasitoids was not significant.”

In discussion: L257-261 “However, this significant pattern was observed only in the NMDS analysis and not in the Mantel test, suggesting that the non-random interactions between hosts and parasitoids could not be simply predicted by their community similarity and associations between the phylogenetic composition of hosts and parasitoids are more complex and require further investigation in the future.”

-- One additional minor point: "site" would be better set as a fixed rather than random term in the linear mixed-effects models, because the site number (2) is too small to make a proper estimate of random component.

Now we treated “site” as a fixed factor in our models, interacting with tree species richness/tree MPD and tree functional diversity to reflect the variation of spatial and tree composition between the two sites. We found the main results did not change, as both sites showed consistent patterns for effects of tree richness/MPD on network metrics, which is more pronounced in one site.

**Public Reviews:**

**Reviewer #1 (Public Review):**
Summary:The authors analyzed how biotic and abiotic factors impact antagonistic host-parasitoid interaction systems in a large BEF experiment. They found the linkage between the tree community and host-parasitoid community from the perspective of the multi-dimensionality of biodiversity. Their results revealed that the structure of the tree community (habitat) and canopy cover influence host-parasitoid compositions and their interaction pattern. This interaction pattern is also determined by phylogenetic associations among species. This paper provides a nice framework for detecting the determinants of network topological structures.Strengths:This study was conducted using a five-year sampling in a well-designed BEF experiment. The effects of the multi-dimensional diversity of tree communities have been well explained in a forest ecosystem with an antagonistic host-parasitoid interaction.The network analysis has been well conducted. The combination of phylogenetic analysis and network analysis is uncommon among similar studies, especially for studies of trophic cascades. Still, this study has discussed the effect of phylogenetic features on interacting networks in depth.Weaknesses:(1) The authors should examine species and interaction completeness in this study to confirm that their sampling efforts are sufficient.(2) The authors only used Rao's Q to assess the functional diversity of tree communities. However, multiple metrics of functional diversity exist (e.g., functional evenness, functional dispersion, and functional divergence). It is better to check the results from other metrics and confirm whether these results further support the authors' results.(3) The authors did not elaborate on which extinction sequence was used in robustness analysis. The authors should consider interaction abundance in calculating robustness. In this case, the author may use another null model for binary networks to get random distributions.(4) The causal relationship between host and parasitoid communities is unclear. Normally, it is easy to understand that host community composition (low trophic level) could influence parasitoid community composition (high trophic level). I suggest using the 'correlation' between host and parasitoid communities unless there is strong evidence of causation.

Thank you very much for your thoughtful and constructive review of our manuscript. We have carefully addressed your comments and made several revisions to improve the clarity and robustness of our work.(1) We appreciate your suggestion regarding species and interaction completeness. To confirm that our sampling efforts were sufficient, we have now included a figure (Fig. S1) showing the species accumulation curve and the coverage of interactions in our study. This ensures that the data collected provide a comprehensive representation of the system. (2) Regarding the use of only Rao’s Q to assess functional diversity, we acknowledge that multiple metrics of functional diversity exist. However, due to the large number of predictors in our analysis, we decided to streamline our approach and focus on Rao’s Q as it provides a robust measure for our research objectives. We have discussed this decision in the revised manuscript and clarified that, while additional metrics could be informative, we believe Rao’s Q sufficiently captures the key aspects of functional diversity in our study. (3) We have elaborated on the robustness analysis and the null model used in our study. Specifically, we now clarified which extinction sequence (random extinction) was used in our manuscript, and explained interaction abundance was incorporated into the robustness calculations (*networklevel function*, weighted=TURE; see L506). (4) We have revised the text to clarify the relationship between host and parasitoid communities. As you correctly pointed out, while it is intuitive that host community composition influences parasitoid community composition, we have reframed our analysis to emphasize the correlation between the two communities rather than implying causation without strong evidence. We have revised the manuscript to reflect this distinction.

**Reviewer #2 (Public Review):**
Summary:In their manuscript, Multi-dimensionality of tree communities structure host-parasitoid networks and their phylogenetic composition, Wang et al. examine the effects of tree diversity and environmental variables on communities of reed-nesting insects and their parasitoids. Additionally, they look for the correlations in community composition and network properties of the two interacting insect guilds. They use a data set collected in a subtropical tree biodiversity experiment over five years of sampling. The authors find that the tree species, functional, and phylogenetic diversity as well as some of the environmental factors have varying impacts on both host and parasitoid communities. Additionally, the communities of the host and parasitoid showed correlations in their structures. Also, the network metrices of the host-parasitoid network showed patterns against environmental variables.Strengths:The main strength of the manuscript lies in the massive long-term data set collected on host-parasitoid interactions. The data provides interesting opportunities to advance our knowledge on the effects of environmental diversity (tree diversity) on the network and community structure of insect hosts and their parasitoids in a relatively poorly known system.Weaknesses:To me, there are no major issues regarding the manuscript, though sometimes I disagree with the interpretation of the results and some of the conclusions might be too far-fetched given the analyses and the results (namely the top-down control in the system). Additionally, the methods section (especially statistics) was lacking some details, but I would not consider it too concerning. Sometimes, the logic of the text could be improved to better support the studied hypotheses throughout the text. Also, the results section cannot be understood as a stand-alone without reading the methods first. The study design and the rationale of the analyses should be described somewhere in the intro or presented with the results.

Thank you very much for your valuable comments and suggestions on our manuscript! We appreciate your feedback and have made revisions accordingly. Specifically, we have rephrased the interpretation of the results and conclusions to better align with the analyses and avoid overstatements, particularly concerning the top-down control in the system. In addition, we have expanded the methods section by providing more details, especially regarding the statistical approaches, to address the points you raised. To enhance the clarity of the manuscript, we have also ensured that the logic of the text better supports the hypotheses throughout. Please see our point-by-point responses below for additional clarifications.

**Recommendations for the authors:**

**Reviewer #1 (Recommendations For The Authors):**
Line 120: "... and large ecosystems susceptible to global change (add citation here)": Citation(s)?

Now we provided the missed citations.

Line 141: Add sampling completeness information.

Now we provide a new figure about sampling completeness (Fig. S1) in the supplementary materials, showing the adequate sampling effort for our study.

Line 151: use more metrics in the evaluation of functional diversity

We used tree functional diversity Rao’s Q, which is an integrated and wildly used metric to represent functional dissimilarity of trees. As our study focus on multiple diversity indices of trees, it would be better to do not pay more attention to one type of diversity. Thank you for your suggestion!

Line 164: host vulnerability. Although generality and vulnerability are commonly used in network analysis, it is better to link these metrics with the trophic level, like the 'host vulnerability' you used. Thus, you can use 'parasitoid generality' instead of 'generality'.

Thanks for your suggestion. Now the metrics were labeled with the trophic levels in the full text.

Line 169: two'.'

Corrected.

Line 173: 'parasitoid robustness' Or 'robustness of parasitoids'?

Now changed it to ‘robustness of parasitoid’.

Lines 173, 468: For the robustness estimations, maybe use null model for binary networks to get random distributions?

Thanks for the suggestion. Actually, we have used Patefield null models to compare the randomized robustness and observed, helping to assess whether the robustness of the observed network is significantly different compared to expected by chance. All robustness indices across plots were significantly different from a random distribution, See results section L197-201.

Line 184: modulating interacting communities of hosts and parasitoids.

Changed accordingly.

Line 186: determined host-parasitoid interaction patterns

Changed accordingly.

Line 191: Biodiversity loss in this study refers to low trophic levels.

Now we clarified this point.

Line 190: understand

Changed accordingly.

Lines 215-216: Reorganize these sentencesLine 227: indirectly influenced by...

Changed accordingly.

Line 238: Be more specific. Which type of further study?

Rephased it more specific.

Lines 297-299: rewrite this sentence to make it more transparent.

Now we rewrote the sentence accordingly.

Line 302: Certain

Changed accordingly.

Line 453: effective

Changed accordingly.

Finally, the authors should check the text carefully to avoid grammatical errors.

Thanks, now we have checked the full text to avoid grammatical errors.

**Reviewer #2 (Recommendations For The Authors):**
I feel that the authors have very interesting data and have a solid set of analyses. I do not have major issues regarding the manuscript, though sometimes I disagree with the interpretation of the results and some of the conclusions might be too far-fetched given the analyses and the results. Additionally, the methods section (especially statistics) was lacking some details, but I would not consider it too concerning at this point.I feel that the largest caveat of the manuscript remains in the representation of the rationale of the study. I felt the introduction could be more concise and be better focused to back up the study questions and hypotheses. Many times, the sentences were too vague and unspecific, and thus, it was difficult to understand what was meant to be said. The authors could mention something more about how community composition of hosts and parasitoids are expected to change with the studied experimental design regarding the metrices you mention in the introduction (stronger hypotheses). The results section cannot be understood as a stand-alone without reading the methods first. The study design and the rationale of the analyses must be described somewhere in the intro or results, if the journal/authors want to keep the methods last structure. Also, the results and discussion could be more focused around the hypotheses. Naturally, these things can be easily fixed.I also disagree with the interpretation of results finding top-down control in the system (it might well be there, but I do not think that the current methods and tests are suitable in finding it). First, the used methodology cannot distinguish parasitoids if the hosts are not there and the probability to detect parasitoid likely depends on the abundance of the host. Thus, the top-down regulation is difficult to prove (is it the parasitoids that have driven the host population down). Secondly, I would be hesitant to say anything about the top-down and bottom-up control in the systems as the data in the manuscript is pooled across five years while the top-down/bottom-up regulation in insect systems usually spans only one season/generation in time (much shorter than five years). Consequently, the analyses are comparing the communities of species that some of most likely do not co-exist (they were found in the same space but not during the same time). Luckily, the top-down/bottom-up effects could potentially be explored by using separately the time steps of the now pooled community data: e.g., does the population of the host decrease in t if the parasitoids are abundant in t-1? There are also other statistical tests to explore these patterns.In the manuscript "Phylogenetic composition" refers to Mean Pairwise Distance. I would use "phylogenetic diversity" instead throughout the text. Also, to my understanding, in trees both "phylogenetic composition" and "phylogenetic diversity" are used even though based on their descriptions, they are the same.Detailed comments:Punctuation needs to be checked and edited at some point (I think copy-pasting had left things in the wrong places). Please check that "-" instead of "-" is used in host-parasitoid.

1-2 The title is not very matching with the content. "Multi-dimensionality" is not mentioned in the text. "phylogenetic composition" -> "phylogenetic diversity"

We didn’t find the role of functional diversity of trees in host-parasitoid interactions, but we still have tree richness and phylogenetic diversity. I also disagree with that using phylogenetic diversity to replace phylogenetic composition, because diversity highlights higher or lower phylogenetic distance among communities, while the later highlights the phylogenetic dissimilarity across communities.

53-57 This sentence is quite vague and because of it, difficult to follow. Consider rephrasing and avoiding unspecified terms such as "tree identity", "genetic diversity", and "overall community composition of higher trophic levels" (at least, I was not sure what taxa/level you meant with them).

Rephased.

L58-61 “Especially, we lack a comprehensive understanding of the ways that biotic factors, including plant richness, overall community phylogenetic and functional composition of consumers, and abiotic factors such as microclimate, determining host–parasitoid network structure and host–parasitoid community dynamics.”

56 I would remove "interact" as no interactions were tested.

Removed accordingly.

59-60 This needs rephrasing. I feel "taxonomic and phylogenetic composition should be just "species composition". To better match, what was done: "taxonomic, phylogenetic, and network composition of both host and parasitoid communities" -> "species and phylogenetic diversity of both host and parasitoid communities and the composition their interaction networks"

Changed accordingly.

62 Remove "tree composition".

Done.

62 Replace "taxonomic" with "species". Throughout the text.

Done.

63-64 "Generally, top-down control was stronger than bottom-up control via phylogenetic association between hosts and parasitoids" I disagree, see my comments elsewhere.

Now we rephased the sentence.

L68-70 “Generally, phylogenetic associations between hosts and parasitoids reflect non-randomly structured interactions between phylogenetic trees of hosts and parasitoids.”

68 "habitat structure and heterogeneity" This is too strong and general of a statement based on the results. You did not really measure habitat structure or heterogeneity.

Now we rephased the statement to avoid strong and general description.

L71-73 “Our study indicates that the composition of higher trophic levels and corresponding interaction networks are determined by plant diversity and canopy cover especially via trophic phylogenetic links in species-rich ecosystems.”

69 Specify "phylogenetic links". Trophic links?

Specified to “trophic phylogenetic links”.

75-77 The sentence is a bit difficult to follow. Consider rephrasing.

Now we rephased it.

L79-82 “Changes in network structure of higher trophic levels usually coincide with variations in their diversity and community, which could be in turn affected by the changes in producers via trophic cascades”

76 Be more specific about what you mean by "community of trophic levels".

Specified to “community composition”.

79 Remove "basal changes of", it only makes the sentence heavier.

Done.

81 What is "species codependence"?

We sim to describe the species co-occurrence depending on their closely relationships. For clarity, now we changed to “species coexistence”

82 What do you mean by "complex dynamics"?

Rephased to “mechanisms on dynamics of networks”.

83 onward: I would not focus so much on top-down/bottom-up as I feel that your current analyses cannot really say anything too strong about these causalities but are rather correlative.

Thanks, we now removed the relevant contents from the discussion. However, we kept one sentence in the Introduction, because it should be highlighted to make reviewers aware of this (the other text on about this were removed).

89 Remove "environmental".

Done.

90 Specify what you mean by "these forces".

Done.

98-99 I have difficulties following the logic here "potential specialization of their hosts may cascade up to impact the parasitoids' presence or absence". Consider rephrasing.

Now we rephased it.

L101-102 “…and their host fluctuations may cascade up to impact the parasitoids’ presence or absence.”

100 Be more specific with "habitat-level changes".

Specified to “community-level changes”

100 I do not see why host-parasitoid systems would be ideal to study "species interactions". There are much simpler and easier systems available.

Changed to “… one of ideal…”

101-103 "influence of" on what?

Now we rephased the sentence.

L104-105 “Previous studies mainly focused on the influence of abiotic factors on host-parasitoid interactions”

104 Be more specific in "the role of multiple components of plant diversity".

Now we specified "the role of multiple components of plant diversity".

L107-108 “…the role of multiple components of plant diversity (i.e. taxonomic, functional and phylogenetic diversity)…”

106 "diversity associations" of what?

“diversity associations between host and parasitoids”.

108 Specify the "direct and indirect effects".

Now we specified it to “…direct and indirect effects (i.e. one pathway and more pathways via other variables)…”

110-113 A bit heavy sentence to follow. Consider rephrasing.

Now we rephased the sentence to make it more readable.

114 Give an example of "phylogenetic dependences".

Done. Phylogenetic dependences (e.g. phylogenetic diversity)

117 Move the "e.g. taxonomic, phylogenetic, functional" within brackets in 117 after "dimensions of biodiversity".

Done.

120 "(add citation here)" Yes please!

Done.

120-121 Specify "such relationships".

Done. Specified to “multiple dimensions of biodiversity”

128-130 This is difficult to follow. Please rephrase.

Now we rephased the sentence.

L135-137 “We aimed to discern the primary components of the diversity and composition of tree communities that affect higher trophic level interactions via quantifying the strength and complexity of associations between hosts and parasitoid.”

131-132 Remove "phylogenetic and". It is redundant to phylogenetic diversity.

Done.

128 Tested robustness does not really capture "stability of associations".

Yes, we agree. Now we rephased the sentence and exclude the “stability” description.

133 Specify "phylogenetic processes".

Now we specified “phylogenetic processes”.

L140-141 “…especially via phylogenetic processes (e.g. lineages of trophic levels diverge and evolve over time)…”

141 I would like to have more details on the data set somewhere in the results. How many individuals and species were found in each plot (on average)? Was there a lot of temporal variation (e.g. between the seasons)? On how many sites were the insect species found?

Thanks for your suggestion. Now we provide more details on the data set in the results (L153-156), including mean values of individuals and species in each plot. However, the temporal variation should be studied for another relative independent topic, as our study focus on the general patter of interactions between hosts and parasitoids. Therefore, we would not put more information on temporal changes to make readers get lost in the text.

153-156 “Among them, we found 56 host species (12 bees and 44 wasps, mean abundance and richness are 400.05 and 45.14, respectively, for each plot) and 50 parasitoid species (38 Hymenoptera and 12 Diptera, mean abundance and richness are 14.07 and 9.05, respectively, for each plot).”

149 tree -> trees

Done.

149 Should there read also some else than "NMDS scores"?

Thanks! Now we provided more details about “NMDS scores”.

L161-162 “(NMDS axis scores; i.e. preserving the rank order of pairwise dissimilarities between samples)”

149 You could mention the amount of variation explained by the first two axes of the NMDSs. Now it is difficult to estimate how much the models actually explain.

Thanks for your comments! However, we could not directly provide the explanatory power of the two axes, because NMDS is based on rank-order distances rather than linear relationships like in PCA. However, the goodness of fit for the NMDS solution is typically evaluated using the stress value. We provide the stress value in the figure caption.

150 "tree MPD" is mentioned for the first time. Spell it out.

Done.

150 Explain "eastness".

Done.

L163-164 *“…eastness (sine-transformed radian values of aspect)”*

151 How was "tree functional diversity" quantified?

Please see methods. L437-L438.

160 Specify that you talk about phylogenetic compositions of the host and parasitoid communities here.

We would keep it refined here, keeping consistent with species composition here. Phylogenetic composition just represents the dissimilarities of phylogenetic linages within a community.

161 Describe "parafit" test here when first mentioned.

Done, see methods L485-487.

182 Keep on referring to tables and figures in the discussion! Also, more clearly discuss your hypotheses. There are lots of discussions on top-down/bottom-up control. It could be good to form a hypothesis on them and predict what kind of patterns would suggest either one and what would you expect to find regarding them.

Now we referred figures and tables in the discussion. As the contents on top-down and bottom-up control were not fit very well with our study (as also suggested by reviewers), so we rephased the discussion and also clearly discuss our hypotheses in the discussion. See L218, L226, and L237 etc.

186 "partly determined host-parasitoid networks" Be more specific.

Done.

L206-207 “…partly determined host-parasitoid network indices, including vulnerability, linkage density, and interaction evenness.”

195 Tell what you mean by "other biotic factors".

Specified it: “…other biotic factors such as elevation and slope…”

197-198 "It seems likely that these results are based on bee linkages to pollen resources" I would be hesitant to conclude this as the bees most likely forage way beyond the borders of the 30m by 30m study plots.

Thanks for your concern about this problem. While it is true that bees can forage beyond 30 x 30m, the study focuses on their nesting behavior and activity within this defined area, rather than their entire foraging range. Existing literature shows bees often forage locally when resources are available (e.g. Ebeling et al., 2012 Oecologia; Guo et al., year, Basic and Applied Ecology). Therefore, we are confident that this pattern could be associated with the resources around the trap nests.

223 "This could be further tested by collecting the food directly used by the wasps (caterpillars)" A bit unnecessary addition.

Thanks for your suggestion. Yes, this definitely is a good point, but currently we don’t have enough data of caterpillars, but we will follow this in the future.

232-238 I disagree with the authors on the interpretation of the causality of the results here. I think that the community of parasitoids simply indicates which host species are available, while the host community does not have an as strong effect on parasitoid community as parasitoids do not utilise the whole species pool of the hosts. (Presence of parasitoid tells that the host is around while the presence of the host does not necessarily tell about the presence of the parasitoid.) To me, this would rather indicate a bottom-up than top-down regulation. Similar patterns are also visible in species communities of hosts and parasites.

Thank you for your suggestion. We agree with you that parasitoids are more depended on hosts, as host could not be always attacked by parasitoids. Now we rephased our explanation to follow this argument.

L254-256 “Such pattern could be further confirmed by the significant association between host phylogenetic composition and parasitoid phylogenetic composition (Fig. 1c), which suggested that their interactions are phylogenetically structured to some extent.”

247-266 The logic in this section is difficult to follow. Try rephrasing.

Now we rephased the section for a clearer logic.

L270-287 “Tree community species richness did not significantly influence the diversity of hosts targeted by parasitoids (parasitoid generality), but caused a significant increase in the diversity of parasitoids per host species (host vulnerability) (Fig. 3a; Table 2). This is likely because niche differentiation often influences network specialization via potential higher resource diversity in plots with higher tree diversity (Lopez-Carretero et al. 2014). Such positive relationship between host vulnerability and tree species richness suggested that host-parasitoid interactions could be driven through bottom-up effects via benefit from tree diversity. For example, parasitoid species increases more than host diversity with increasing tree species richness (Guo et al. 2021), resulting increasing of host vulnerability at community level. According to the enemies hypothesis (Root 1973), which posits a positive effects of plant richness on natural enemies, the higher trophic levels in our study (e.g. predators and parasitoids) would benefit from tree diversity and regulate herbivores thereby (Staab and Schuldt 2020). Indeed, previous studies at the same site found that bee parasitoid richness and abundance were positively related to tree species richness, but not their bee hosts (Fornoff et al. 2021, Guo et al. 2021). Because our dataset considered all hosts and reflects an overall pattern of host-parasitoid interactions, the effects of tree species richness on parasitoid generality might be more complex and difficult to predict, as we found that neither tree species richness nor tree MPD were related to parasitoid generality.”

249 "This is likely because niche differentiation often influences network specialization via potential higher resource diversity in plots with higher tree diversity" This is a bit contradicting your vulnerability results as niche differentiation should increase specialization and diversity and specialization should decrease vulnerability (less host per parasitoid).

Thanks! We understand that the concepts of “generality” and “vulnerability” can be a bit confusing. To clarify, “fewer hosts per parasitoid” actually corresponds to lower generality at the community level.

332-337 How did you select the species growing on your plots? Or was only species number considered? What was the pool of tree species growing on the selected plots? Was the selection similar at both sites?

Now we provided more information on the experiment design.

L354-356 *“The species pools of the two plots are nonoverlapping (16 species for each site). The composition of tree species within the study plots is based on a “broken-stick” design (see Bruelheide et al. 2014).”*

342 Remove "centrally per plot"?

Done.

346-347 Was the selection of different reed diameters similar in all the plots?

Diameters and the relative distribution of diameters was similar in all trap nests.

399 & 432 Are "phylogenetic diversity of the tree communities" and "phylogenetic composition of trees" the same? They are both described as mean pairwise distance.

These two are actually different, as we use this to distinguish the phylogenetic diversity with communities and rank order of dissimilarities between tree communities. Here, the phylogenetic diversity of the tree communities is mean pairwise phylogenetic distance of species for tree communities. Tree phylogenetic composition is the rank order of pairwise dissimilarities between tree communities based on NMDS.

400 Do you think that MPD makes any sense with the monocultures (value is always 0)? Does this have a potential to bias your analyses and result?

We agree your point. However, we do not think that this is a major problem in the analyses. We followed the experimental design and considered low phylogenetic relatedness of tree species in a plot (Likewise in monocultures, the tree species richness is always 1).

402-405 MNTD is not mentioned before or after this. Consider removing this section.

We tested the potential effects of MNTD in our models. Now we mentioned it in our results.

L194-195 “Tree mean nearest taxon distance (MNTD) was unrelated to any network indices.”

405 "Phylogenetic metrics of trees" Which ones?

Both tree MPD and MNTD. Now we have noted it in the manuscript. (L432)

410 Further details on "Rao's Q" and how the functional diversity of the communities was calculated are needed.

Now more details were provided.

L435-438 “Specifically, seven leaf traits were used for calculation of tree functional diversity, which was calculated as the mean pairwise distance in trait values among tree species, weighted by tree wood volume, and expressed as Rao's Q”

413 Specify "higher trophic levels".

Now we specified the trophic levels.

L440-441 “…higher trophic levels in our study area, such as herbivores and predators”

417-424 What about the position of the plots within study sites? Is there potential for edge effects (e.g. bees finding easier the trap nest close to the edge of the experimental forest)? Were there any differences between the two sites? What is the elevation range of the plots?

Thanks for concerning the details of our study. First, all the plots were randomly distributed within the study sites (see Fig. S2). Admittedly, there are several plots are located in the edges of the site. However, we did not consider the potential edge effects in our analysis. Of course, this will be a good point in our future studies. Moreover, the biggest difference between the two is the non-overlapping tree species pool, and the two study sites are apart from 5 km in the same town. Finally, there is not too distinct elevation gradient across the plots (112 m - 260 m).

432-434 "The species and phylogenetic composition of trees, hosts, and parasitoids were quantified at each plot with nonmetric multidimensional scaling (NMDS) analysis based on Morisita-Horn distances" This section needs to be more specific and detailed. Did you do the NMDS separately for each plot as suggested in the text?

We provided more details of the section.

L462-465 “The minimum number of required dimensions in the NMDS based on the reduction in stress value was determined in the analysis (k = 2 in our case). We centred the results to acquire maximum variance on the first dimension, and used the principal components rotation in the analysis.”

435 Specify how picante was used (function and arguments)!

Now we specified the function.

L465-467 “The phylogenetic composition was calculated by mean pairwise distance among the host or parasitoid communities per plot with the R package “picante” with ‘mpd’ function.”

436 "standardized values" Of what? How was the standardisation done?

Now we citied a supplementary table (Table S2) to specify it (see L469). For the standardization, we used *‘scale’* function in R, which standardizes data by centering and scaling data. Specifically, it subtracts the mean and divides by the standard deviation for each variable.

443 Provide more details on parafit.

Actually, we have provided the reason why we use the parafit test and the usage.

L483-486 “We used a parafit test (9,999 permutations) with the R package “ape” to test whether the associations were non-random between hosts and parasitoids. This is widely used to assess host-parasite co-phylogeny by analyzing the congruence between host and parasite phylogenies using a distance-based matrix approach.”

449-451 Rephrase the sentence.

Rephased.

L490-491 “We constructed quantitative host-parasitoid networks at community level with the R package “bipartite” for each plot of the two sites.”

451 "six" Should this be five?

Yes, should be five, thanks.

470-481 What package and function were used for the LMMs?

As we now used linear models, we do no longer use a R package for LMMs.

470 "mix" -> mixed

Changed to linear models.

472 "six" Should this be five?

Again, we changed it to five.

479-481 How did you treat the variables from the two different sites when testing for the correlations to avoid two geographic clusters of data points?

Now we considered the two study sites as fixed factor in our linear models. Moreover, tree-based variables were additionally included as interaction terms with the study sites.

501 "mix" -> mixed

Changed to linear models.

The panel selection for figures 3 and 4 seems random. Justify it!

Thank you. To avoid including too many figures in the main text, which could potentially confuse readers, we have selected the key results that are of primary interest. The remaining figures are provided in the appendix for reference.

533 "Note that axes are on a log scale for tree species richness." Why the log-scale if the analyses were performed with linear fit? Also, the drawn regression lines do not match the model description (non-linear, while a linear model is described in the text). The models should probably be described in more detail.

We used log-transformed to promote the normality of the data. The drawn regression lines are linear lines, which fit our models.

539 "Values were adjusted for covariates of the final regression model." How?

We used residual plot to directly visualizes the relationship between the predictor and the response variable with the fitted regression line, making it easier to assess the model's fit.

Fig. S4 text does not match the figure.

Thanks! We now deleted the unmatched text in the figure.